# SDDuSR: Sparse Feature Matching and Token Dictionary Learning for Dual-Lens Super-Resolution

## Abstract

Dual-Lens Super-Resolution (DuSR) is an application of Reference-based image Super-Resolution (RefSR) in real-world scenarios. Unlike RefSR, DuSR uses the telephoto image as the high-resolution reference image (Ref) and the wide-angle image as the low-resolution image (LR), where LR and Ref share the field of view (FoV) within a certain area. Then, the Ref image is used to assist the LR image in super-resolution. The existing DuSR methods all employ dense feature matching and warping operation to identify and transfer the high-resolution features of the Ref image to the LR image. However, this approach has two key issues: (1) the smooth low-frequency regions in the LR image can achieve good visual effects without any reference, which leads to significant computational redundancy caused by dense feature matching, and (2) due to the inherent limitations of the warping operation, it is not possible to fully utilize the high-resolution features of the Ref image. To address these issues, we propose a DuSR method based on Sparse Feature Matching and Token Dictionary Learning, called SDDuSR. Specifically, we introduce a mask generator to separate the high-frequency regions from the low-frequency regions of the image, and perform feature matching only on the high-frequency regions, which significantly reduces the computational load during the feature matching stage. Moreover, to fully utilize the features of the Ref image, we abstract it into a token dictionary and employ a dictionary learning strategy to assist the LR image in super-resolution. Extensive experiments have demonstrated that our method achieves state-of-the-arts (SOTA) performance in both quantitative and qualitative aspects.

## 1 Introduction

Single Image Super-Resolution (SISR) (Dong et al., 2015; Liang et al., 2021) aims to reconstruct a degraded low-resolution (LR) image into a high-resolution (HR) image. However, due to the limited information available from a single image, SISR cannot reconstruct the rich details in the image when the degradation is severe. To address this issue, Reference-based Image Super-Resolution (RefSR) (Zhang et al., 2019; Jiang et al., 2021) aims to introduce an additional high-resolution reference image (Ref) that is similar to the LR image to assist the LR image in super-resolution, thereby achieving better visual effects. Although RefSR can achieve better results, it is difficult to find a suitable Ref image for each LR image in real-world scenarios.

Fortunately, with the development and widespread adoption of smartphones, we can more easily obtain pairs of images with different resolutions. The images captured by the wide-angle lenses of smartphones have lower resolution and a larger field of view (FoV), while the images captured by the telephoto lenses have higher resolution and a smaller FoV. Moreover, the telephoto image and the wide-angle image have an overlapping FoV. We refer to the region in the LR image that has the same FoV as the Ref image as the center region, and the rest of the areas as the corner regions. DCSR (Wang et al., 2021) uses the wide-angle image as the LR image and the telephoto image as the Ref image, as shown in Figure 1, and first proposes the Dual-Lens Super-Resolution (DuSR). For both RefSR and DuSR, one of the key issues is how to find features in the Ref image that are similar to those in the LR image and fully utilize these features to assist super-resolution.

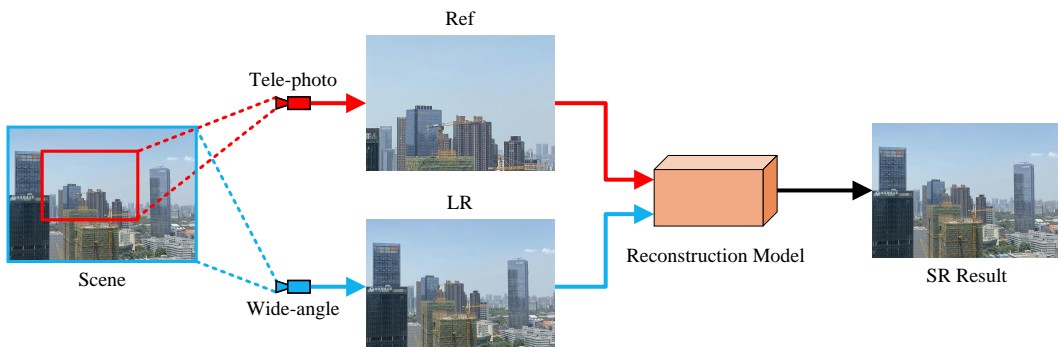

Figure 1: A demonstration of the Dual-Lens image acquisition and DuSR processes. The red rectangular box indicates the overlapping FoV area (center region), and the rest indicates the corner region. DuSR uses the telephoto lens as a reference to assist the wide-angle lens in super-resolution.

Currently, most methods (Yang et al., 2020; Zhang et al., 2022b; Yue et al., 2024) identify and transfer the reference features with the highest similarity through dense feature matching and warping operations. Dense feature matching first divides the LR features and Ref features into patches and calculates the normalized inner product between patches as the similarity score. Finally, based on the index of the Ref patch with the highest similarity, the Ref features are warped and then fused with the LR features. Although this method can to some extent find appropriate reference features, it is also confronted with two key issues: (1) the smooth low-frequency regions in the LR image can achieve good visual effects without any reference, which leads to significant computational redundancy caused by dense feature matching, and (2) due to the inherent limitations of the warping operation, it is not possible to fully utilize the high-resolution features of the Ref image.

Regarding the first issue, we find that not all LR features and Ref features need to have their similarity computed. For example, the smooth low-frequency regions in the LR image can achieve good results solely through the reconstruction capability of the model, without requiring any reference. Similarly, the low-frequency regions in the Ref image also do not need to be involved in the feature matching process. Therefore, dense feature matching leads to significant computational redundancy. Regarding the second issue, as the warping operation is a rigid process, it is essentially a patch reassembly operation. The warping operation reassembles the Ref patch with the highest similarity corresponding to each LR patch into a new feature map. When LR patches in certain areas (for example, the corner region), cannot find reference patches with high similarity, relying solely on patch reassembly cannot achieve satisfactory results. In this case, forcibly using warping operations will introduce Ref features with low similarity, resulting in blurring and artifacts.

To address the aforementioned issues, we propose a DuSR method based on Sparse Feature Matching (SFM) and Token Dictionary Learning (TDL), called SDDuSR. Specifically, we first introduce a mask generator to separate the high-frequency and low-frequency regions of the LR image and the Ref image. During the feature matching stage, similarity is only computed for the features in the high-frequency regions, effectively reducing the computational load in this phase. We can allocate the calculation amount reduced by SFM to the TDL strategy. Secondly, inspired by the use of dictionaries to represent image features (Yang et al., 2010; Zhang et al., 2024), we introduce the TDL strategy to fully exploit the Ref features and avoid the limitations of the warping operation. TDL abstracts the feature maps into higher-level dictionary features through a token dictionary, where each token in the dictionary can represent different types of features. Specifically, TDL consists of two phases: updating and learning. In the updating phase, we abstract the Ref features into dictionary features and update them into the predefined token dictionary. In the learning phase, the LR features acquire useful high-resolution features from the token dictionary through cross-attention. During the training process, TDL gradually abstracts the Ref features of the entire dataset as external priors. When the features of certain regions in the LR image cannot be well referenced from the current Ref image or there are no similar features in the Ref image, TDL can better enrich the details of these regions by looking them up in a dictionary. Through the combination of SFM and TDL, SDDuSR can significantly improve the performance when the overall computing load is almost unchanged

In summary, the contributions of this paper are as follows:

- We propose a Sparse Feature Matching strategy that separates the high-frequency and low-frequency regions of the image. This approach reduces the computational complexity in the feature matching stage while avoiding significant performance degradation.

- We propose a Token Dictionary Learning strategy, which updates the high-resolution Ref features into the token dictionary and then interacts the LR features with the token dictionary through cross-attention. This strategy effectively avoids the limitations of the warping operation, thereby more fully exploiting the Ref features.

- We conducted both quantitative and qualitative experiments on multiple DuSR datasets. The experiments demonstrate that our method can effectively reduce the computational load in the feature matching stage, while achieving state-of-the-arts (SOTA) performance.

## 2 RELATED WORK

### 2.1 REFERENCE-BASED IMAGE SUPER-RESOLUTION

RefSR utilizes additional high-resolution images (HR) as references to enhance the super-resolution performance. Existing RefSR methods transfer features from the reference image through spatial alignment or dense feature matching strategies. CorssNet (Zheng et al., 2018) and SSEN (Shim et al., 2020) respectively align the entire Ref feature with the LR feature using optical flow and deformable convolution (DCN) (Dai et al., 2017). However, these spatial alignment-based methods struggle with the issue of large feature offsets that cannot be resolved effectively. SRNTT (Zhang et al., 2019) divides the Ref features and LR features into patches and transfers the reference features by calculating the similarity between patches through dense feature matching. TTSR (Yang et al., 2020) introduces the Transformer (Vaswani et al., 2017) structure into RefSR, combining soft attention and hard attention to better transfer reference features. To address the resolution gap and scale transformation issues between LR features and Ref features, $C^2$-Matching (Jiang et al., 2021) further improves the accuracy of feature matching through contrastive learning and knowledge distillation (Gao et al., 2018; Liu et al., 2019) strategies. MASA (Lu et al., 2021) and AMSA (Xia et al., 2022) have optimized the feature matching process and further proposed new feature fusion strategies. RRSR (Zhang et al., 2022a) introduces the reciprocal learning strategy and weight generating networks into RefSR. DATSR (Cao et al., 2022) uses U-Net (Ronneberger et al., 2015) and SwinIR (Liang et al., 2021) into RefSR, further enhancing the model's feature representation capabilities.

### 2.2 DUAL-LENS SUPER-RESOLUTION

Although RefSR has achieved good results, it is currently trained based on the synthetic dataset CUFED5 (Zhang et al., 2019), and it is difficult to obtain appropriate high-resolution reference images in real-world scenarios. Compared with RefSR, DuSR is more practical because we can easily obtain telephoto images and wide-angle images through the multi-camera systems of smartphones. Since wide-angle images sacrifice resolution to achieve a larger FoV, telephoto images can be directly used as references for wide-angle images. DCSR (Wang et al., 2021) is the first to introduce RefSR into DuSR. Since the captured wide-angle images lack labels, DCSR first trains using downsampled synthetic data and then fine-tunes through an Self-supervised Real-image Adaptation (SRA) strategy to adapt to real-world scenarios. SelfDZSR (Zhang et al., 2022b) proposes a self-supervised learning method that directly uses the areas with overlapping FoV in the wide-angle images as LR and the telephoto images as HR for end-to-end training. ZeDuSR (Xu et al., 2023) employs contrastive learning and a discriminator network (Goodfellow et al., 2020) to crop corresponding image pairs (LR, HR) from wide-angle and telephoto images, and then conducts SISR training. Due to the resolution gap between telephoto images and wide-angle images, the accuracy of feature matching can be affected. KeDuSR (Yue et al., 2024) proposes a kernel-free matching strategy. KeDuSR performs dense feature matching between the wide-angle image and its center region, and then transfers features from the telephoto image based on the matching results. Moreover, KeDuSR (Yue et al., 2024) proposes three DuSR datasets with complete triplets (LR, Ref, HR), which enable end-to-end training.

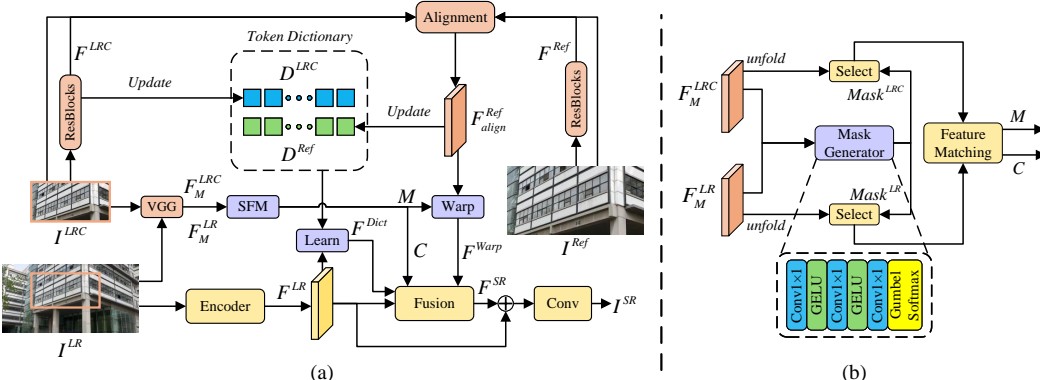

Figure 2: (a) The architecture of SDDuSR. We perform SFM between $I^{LR}$ and $I^{LRC}$ to obtain the index map $M$ and the confidence map $C$, and then warp the Ref feature $F_{align}^{Ref}$ to obtain the feature $F^{Warp}$. To avoid the resolution gap, we define two token dictionaries $D^{LR}$ and $D^{Ref}$. TDL first updates $F^{LRC}$ and $F_{align}^{Ref}$ into two token dictionaries, and then $F^{LR}$ interacts with the token dictionaries to obtain the dictionary features $F^{Dict}$. (b) The inference phase of SFM. SFM inputs $F_M^{LR}$ and $F_M^{LRC}$ into the mask generator to produce masks $Mask^{LR}$ and $Mask^{LRC}$. Then, based on the positions where the mask is 1, the corresponding patches are selected for feature matching.

## 3 METHODOLOGY

### 3.1 THE OVERALL ARCHITECTURE

In DuSR, we use the wide-angle image as the LR image and the telephoto image as the Ref image, denoted as $I^{LR}$ and $I^{Ref}$, respectively. The overall architecture of SDDuSR is shown in Figure 2(a). SDDuSR first extracts the features of the LR image through an encoder, and then performs initial upsampling (Shi et al., 2016) to obtain $F^{LR}$. The encoder is composed of several residual blocks (He et al., 2016) that incorporate channel attention (Hu et al., 2018). Due to the resolution gap between $I^{LR}$ and $I^{Ref}$, direct feature matching between them would result in inaccurate identification of the Ref features. Since the center region $I^{LRC}$ of the LR image shares the FoV with $I^{Ref}$, to avoid the resolution gap, we perform SFM between $I^{LR}$ and $I^{LRC}$ (Yue et al., 2024) to obtain the index map $M$ and the confidence map $C$. However, despite $I^{LRC}$ and $I^{Ref}$ sharing the same FoV, there is still slight misalignment between them. Therefore, we use several residual blocks to extract the features of $I^{LRC}$ (denoted as $F^{LRC}$) and the features of $I^{Ref}$ (denoted as $F^{Ref}$). Then, we feed the feature maps along with the original images into an alignment module to obtain the aligned Ref features $F_{align}^{Ref}$. The alignment module is composed of the optical flow estimation network SpyNet (Ranjan & Black, 2017) and DCN (Dai et al., 2017). Based on the index map $M$ obtained from SFM, we warp $F_{align}^{Ref}$ to obtain the high-resolution Ref feature $F^{Warp}$.

Since feature matching and warping operations alone are not sufficient to fully exploit the Ref features, we introduce the TDL strategy to further assist $I^{LR}$ in super-resolution. Similarly, to avoid the resolution gap, TDL defines two token dictionaries $D^{LRC}$ and $D^{Ref}$ to represent features at different resolutions. In the updating phase, the features of $F^{LRC}$ and $F_{align}^{Ref}$ are updated into $D^{LRC}$ and $D^{Ref}$ through cross-attention. Subsequently, $F^{LR}$ also interacts with two token dictionaries through cross-attention to obtain the high-resolution feature $F^{Dict}$. Then, we fuse $F^{LR}$, $F^{Warp}$, and $F^{Dict}$ obtained above to get $F^{SR}$, the fusion process is formulated as

$$F^{SR} = Fusion(concat(F^{LR}, h(C) \cdot F^{Warp}, F^{Dict})), \tag{1}$$

where $h(\cdot)$ denotes the learnable convolutional layer, and $C$ represents the confidence map. Finally, the number of channels is adjusted through a convolutional layer to reconstruct the final result $I^{SR}$. The fusion module is composed of several residual blocks that incorporate both channel and spatial attention (Woo et al., 2018).

## 3.2 Sparse Feature Matching (SFM)

In $I^{LR}$ and $I^{LRC}$, the low-frequency regions of the image do not require feature matching, as the model's own learning capability can effectively reconstruct the features of the low-frequency regions. Inspired by DynamicVit (Rao et al., 2021) and CAMixer (Wang et al., 2024), we introduce a mask generator to separate the low-frequency and high-frequency regions of the image. We first map $I^{LR}$ and $I^{LRC}$ to the feature space using VGG (Simonyan & Zisserman, 2014) to obtain the feature maps $F_M^{LR}$ and $F_M^{LRC}$. Dense feature matching typically obtains the index map $M$ and the confidence map $C$ through *argmax* and *max* operations after computing the similarity scores between them. However, the *argmax* operation cannot backpropagate gradients, and the *max* operation only propagates gradients at the position of the maximum value, which prevents the mask generator from being fully trained. Therefore, the training phase and inference phase of SFM adopt different processing methods, with the inference phase shown in Figure 2(b).

**Training Phase.** To better demonstrate that dense feature matching indeed introduces computational redundancy, during the training phase, we only pass $F_M^{LR}$ through the mask generator to separate the high-frequency and low-frequency regions. During the inference phase, we process both $F_M^{LR}$ and $F_M^{LRC}$ simultaneously.

During the training phase, we first conduct conventional dense matching. For the input $F_M^{LR}$ and $F_M^{LRC}$, we divide them into $3 \times 3$ patches $P_i^{LR} \in \mathbb{R}^{M_{LR} \times d}$ and $P_i^{LRC} \in \mathbb{R}^{M_{LRC} \times d}$ through the $unfold$ operation (with a stride and padding of 1), where $M_{LR} = H_{LR} \times W_{LR}$ and $M_{LRC} = H_{LRC} \times W_{LRC}$ represent the number of patches, and $d$ denotes the dimension of the patches. Then, we calculate the normalized inner product between patches to obtain the similarity score $S_{i,j} \in \mathbb{R}^{M_{LRC} \times M_{LR}}$ as

$$S_{i,j} = \left\langle \frac{P_i^{LRC}}{\|P_i^{LRC}\|}, \frac{P_j^{LR}}{\|P_j^{LR}\|} \right\rangle. \tag{2}$$

To achieve the same effect as *argmax* and *max* operations while ensuring the complete propagation of gradients, we transform these operations into the form of matrix computations through the Straight-Through Estimator (STE) strategy (Bengio et al., 2013). Specifically, we first apply the softmax operation along the $i$-axis to $S_{i,j}$ to obtain $S_{i,j}^{softmax}$, and then convert it into a one-hot form $S_{i,j}^{onehot}$, where the position of $P_i^{LRC}$ with the highest similarity to $P_j^{LR}$ is set to 1. We use $S_{i,j}^{onehot}$ for forward propagation and $S_{i,j}^{softmax}$ for backward propagation through the STE strategy, thereby bypassing the issue of gradient non-propagation during the one-hot conversion process. This process is formulated as

$$M_{i,j}^{onehot} = d(S_{i,j}^{onehot}) + S_{i,j}^{softmax} - d(S_{i,j}^{softmax}), \tag{3}$$

where $M_{i,j}^{onehot} \in \mathbb{R}^{M_{LRC} \times M_{LR}}$ represents the one-hot index map, and $d(\cdot)$ denotes the *detach* operation in PyTorch for gradient truncation.

To obtain the confidence map, we calculate the Hadamard product between $S_{i,j}$ and $M_{i,j}^{onehot}$, and then sum along the $i$-axis to get the confidence map $C^{dense} \in \mathbb{R}^{M_{LR}}$. This process is formulated as

$$C^{dense} = \sum_i (S_{i,j} \cdot M_{i,j}^{onehot}). \tag{4}$$

Then, we input $F_M^{LR}$ into the mask generator and obtain a binary mask $Mask^{LR}$ through Gumbel-Softmax(Jang et al., 2016), where the high-frequency regions are marked as 1 and the low-frequency regions are marked as 0. After obtaining the mask $Mask^{LR}$, we divide $F_{align}^{Ref}$ into patches $P_k^{Ref} \in \mathbb{R}^{\hat{d} \times M_{LRC}}$, then perform matrix multiplication between $P_k^{Ref}$ and $M_{i,j}^{onehot}$ to achieve the same effect as the warping operation. Finally, the resulting output is subjected to a Hadamard product with $Mask^{LR}$, marking the positions that do not need to participate in feature matching as 0. Subsequently, the patches are restored to the feature map form through the $fold$ operation. For the confidence map $C^{dense}$, we also conduct Hadamard product with $Mask^{LR}$ to mark the confidence of the positions that do not need matching as 0. These processes are formulated as

$$F^{Warp} = fold(P_k^{Ref} M_{i,j}^{onehot} \cdot Mask^{LR}), \quad C = R(C^{dense} \cdot Mask^{LR}) \tag{5}$$

where $R$ represents the reshape operation and both $F^{Warp}$ and $C$ are marked as 0 at the positions that do not need matching. Through the above approach, we achieve the same effect as the *argmax* and *max* operations while avoiding the issue of gradient non-propagation.

**Inference Phase**. Due to the gradient propagation issue during the training phase, we multiply the results of dense feature matching with the mask to achieve the purpose of SFM, but the computational load does not decrease. Therefore, during the inference phase, we input $F_M^{LR}$ and $F_M^{LRC}$ into the mask generator simultaneously to generate masks $Mask^{LR}$ and $Mask^{LRC}$. Then, based on the positions where the values are 1 in both masks, the corresponding $P_i^{LR}$ and $P_j^{LRC}$ are selected to calculate the similarity score $S_{i,j}$ according to Equation 2. Patches at positions where the mask is 0 do not participate in the calculation of similarity scores. Finally, the index map $M$ and the confidence map $C$ are obtained using $argmax_j S_{i,j}$ and $max_j S_{i,j}$ operations, and then $F^{Warp}$ is obtained by warping $P_k^{Ref}$. For $C$ and $F^{Warp}$, we set the values of the positions that did not participate in feature matching to 0.

### 3.3 Token Dictionary Learning (TDL)

Due to the rigidity limitations of warping operations, satisfactory reference features cannot be obtained solely through patch reassembly. Inspired by the use of dictionaries to represent image features (Yang et al., 2010; Zhang et al., 2024), we introduce TDL to more fully exploit the Ref features. TDL consists of an updating phase and a learning phase. During the training process, TDL can abstract features from feature maps into a dictionary, where each token can represent different types of features.

In the updating phase, we update features $F^{LRC}$ and $F_{align}^{Ref}$ into the token dictionaries through cross-attention. Specifically, to avoid the resolution gap, we first define two token dictionaries $D^{LRC} \in \mathbb{R}^{N \times d}$ and $D^{Ref} \in \mathbb{R}^{N \times d}$, where $N$ and $d$ represent the number and dimension of the tokens, respectively. Subsequently, we use fully connected layers to generate $Q_{up}$ from $D^{LRC}$, generate $K_{up}$ and $V_{up}^{LRC}$ from $F^{LRC}$, and generate $V_{up}^{Ref}$ from $F_{align}^{Ref}$. Since $F^{LRC}$ and $F_{align}^{Ref}$ have the same FoV but different resolutions, we compute cross-attention between $Q_{up}$ and $K_{up}$, and then update $D^{LRC}$ and $D^{Ref}$ simultaneously. We set $N \ll H^{LRC}W^{LRC}$ to maintain a low computational cost. These process are formulated as

$$A_{up} = SoftMax(Q_{up}K_{up}^T), \tag{6}$$

$$\hat{D}^{LRC} = A_{up}V_{up}^{LRC}, \hat{D}^{Ref} = A_{up}V_{up}^{Ref}, \tag{7}$$

$$D^{LRC} = s D^{LRC} + (1-s)\hat{D}^{LRC}, \tag{8}$$

$$D^{Ref} = s D^{Ref} + (1-s)\hat{D}^{Ref}, \tag{9}$$

where $\hat{D}^{LRC}$ and $\hat{D}^{Ref}$ represent the content to be updated into the token dictionaries, and $s$ is a learnable parameter with a value range between 0 and 1.

In the learning phase, we generate $Q_L$ from $F^{LR}$, generate $K_L$ from $D^{LRC}$, and generate $V_L$ from $D^{Ref}$. Since both $F^{LR}$ and $D^{LRC}$ represent low-resolution features, computing cross-attention between $F^{LR}$ and $D^{LRC}$ can effectively avoid the resolution gap. Finally, the attention map is matrix-multiplied with $V_L$ to obtain the final high-resolution dictionary feature $F^{Dict}$. These process are formulated as

$$A_L = SoftMax(Q_L K_L^T), \tag{10}$$

$$F^{Dict} = A_L V_L. \tag{11}$$

### 3.4 Loss Functions

Like previous RefSR methods, we also use reconstruction loss $L_{rec}$ (Lai et al., 2017), perceptual loss $L_{per}$ (Johnson et al., 2016), and adversarial loss $L_{adv}$ (Jolicoeur-Martineau, 2018; Goodfellow et al., 2020) for training.

Moreover, if no constraints are imposed on the generated masks, the model will learn to set all values in the masks to 1 to achieve the best performance. Therefore, during training, we perform $\sum_C |HR - HR \downarrow\uparrow|$ on the HR image to represent the richness of details for each pixel, where

Table 1: Quantitative comparisons on DuSR-Real, CameraFusion-Real and RealMCVSR-Real. Bold and underlined indicate the best and second-best performance, respectively. The suffix '-rec' means only reconstruction loss and mask loss are used for training.

| Method | DuSR-Real | | CameraFusion-Real | | RealMCVSR-Real | |
|---|---|---|---|---|---|---|
| | Full-Image | Corner-Image | Full-Image | Corner-Image | Full-Image | Corner-Image |
| | PSNR / SSIM / LPIPS | PSNR / SSIM | PSNR / SSIM / LPIPS | PSNR / SSIM | PSNR / SSIM / LPIPS | PSNR / SSIM |
| RCAN-rec | 26.44 / 0.8676 / 0.147 | 26.33 / 0.8667 | 25.67 / 0.8049 / 0.308 | 25.45 / 0.8012 | 25.96 / 0.8033 / 0.234 | 26.12 / 0.8065 |
| SwinIR-rec | 26.14 / 0.8601 / 0.157 | 26.11 / 0.8597 | 25.32 / 0.8007 / 0.315 | 25.22 / 0.7985 | 25.78 / 0.7982 / 0.246 | 25.94 / 0.8015 |
| ESRGAN | 25.78 / 0.8622 / 0.152 | 25.77 / 0.8617 | - | - | - | - |
| BSRGAN | 24.77 / 0.8227 / 0.202 | 24.71 / 0.8225 | - | - | - | - |
| TTSR-rec | 26.48 / 0.8676 / 0.147 | 26.17 / 0.8631 | 25.83 / 0.8044 / 0.311 | 25.62 / 0.7996 | 25.92 / 0.8017 / 0.235 | 25.98 / 0.8036 |
| MASA-rec | 26.36 / 0.8592 / 0.160 | 26.25 / 0.8582 | 25.78 / 0.8030 / 0.303 | 25.58 / 0.7988 | 25.95 / 0.7989 / 0.239 | 26.07 / 0.8020 |
| DATSR-rec | 26.17 / 0.8583 / 0.157 | 26.11 / 0.8579 | - | - | 25.81 / 0.7975 / 0.242 | 25.95 / 0.8007 |
| DCSR-rec | 26.77 / 0.8748 / 0.134 | 26.29 / 0.8635 | 26.02 / 0.8123 / 0.293 | 25.51 / 0.8016 | 26.28 / 0.8111 / 0.217 | 26.08 / 0.8048 |
| DCSR | 26.19 / 0.8553 / 0.110 | 25.75 / 0.8425 | 25.47 / 0.7605 / 0.165 | 25.08 / 0.7512 | 25.85 / 0.7966 / 0.186 | 25.58 / 0.7793 |
| SelfDZSR-rec | 26.27 / 0.8559 / 0.158 | 26.10 / 0.8548 | 25.94 / 0.8041 / 0.283 | 25.68 / 0.8005 | 25.33 / 0.7928 / 0.246 | 25.30 / 0.7952 |
| SelfDZSR | 25.98 / 0.8455 / 0.105 | 25.81 / 0.8442 | 25.64 / 0.7790 / 0.151 | 25.39 / 0.7753 | 25.24 / 0.7786 / 0.175 | 25.23 / 0.7805 |
| ZeDuSR-rec | 25.41 / 0.8247 / 0.191 | 25.21 / 0.8216 | 26.16 / 0.7920 / 0.279 | 25.87 / 0.7871 | 24.98 / 0.7702 / 0.262 | 24.93 / 0.7720 |
| KeDuSR-rec | 27.66 / **0.8890** / 0.117 | 27.24 / 0.8750 | 27.53 / **0.8292** / 0.276 | 26.93 / 0.8169 | 27.05 / **0.8406** / 0.180 | 26.56 / **0.8139** |
| KeDuSR | 27.18 / 0.8752 / **0.084** | 26.77 / 0.8593 | 27.00 / 0.7931 / **0.133** | 26.43 / 0.7768 | 26.42 / 0.8184 / **0.127** | 25.95 / 0.7875 |
| SDDuSR-rec | **27.81** / 0.8874 / 0.121 | **27.41** / **0.8752** | **27.60** / 0.8274 / 0.280 | **27.06** / **0.8174** | **27.09** / 0.8386 / 0.183 | **26.61** / 0.8131 |
| SDDuSR | 27.20 / 0.8722 / **0.084** | 26.79 / 0.8568 | 27.02 / 0.7897 / 0.135 | 26.49 / 0.7759 | 26.71 / 0.8290 / 0.151 | 26.27 / 0.8038 |

Table 2: Ablation study on SFM and TDL (the left side) and the number of tokens $N$ (the right side).

| Baseline | SFM | TDL | PSNR | $N$ | PSNR |
|---|---|---|---|---|---|
| ✓ | × | × | 27.66 | 64 | 27.75 |
| ✓ | ✓ | × | 27.65 | 128 | **27.81** |
| ✓ | ✓ | ✓ | **27.81** | 192 | 27.76 |
| - | - | - | - | 256 | 27.76 |

Table 3: Computational complexity analysis between Dense Feature Matching (DFM) and SFM. The unit of computational complexity is represented by GFLOPS.

| Method | DuSR-Real | CameraFusion-Real | RealMCVSR-Real |
|---|---|---|---|
| DFM | 319 | 10440 | 319 |
| SFM | 222 (↓ 30%) | 7079 (↓ 32%) | 229 (↓ 28%) |

$C$ represents the channel dimension, ↓ and ↑ represent downsampling and upsampling operations, respectively. Subsequently, we sort it and take the median value as the threshold, setting the parts greater than the threshold to 1 and those less than the threshold to 0 to generate the mask label $Mask_{label}$. Finally, we calculate the Mean Squared Error (MSE) between $Mask^{LR}$ and $Mask_{label}$ as the mask loss $L_{mask}$.

In summary, the final loss function can be expressed as:

$$L = L_{rec} + \lambda_1 L_{mask} + \lambda_2 L_{per} + \lambda_3 L_{adv}, \tag{12}$$

where the weight parameters $\lambda_1$, $\lambda_2$, and $\lambda_3$ are equal to $1 \times 10^{-3}$, $1 \times 10^{-3}$, and $1 \times 10^{-4}$, respectively. Since we hope the mask generator to be more dominated by the reconstruction loss rather than the mask loss during the learning phase, we only use $L_{mask}$ as a constraint term and set its weight to a small value. Please note that in the loss functions, $L_{mask}$ is only used as a lower bound to avoid the model learning all the values in the mask as 1. The proportion of the sparse part of the final mask does not strictly correspond to $L_{mask}$. We provide two training results: one that uses only reconstruction loss and mask loss, and the other that uses all losses.

## 4 EXPERIMENTAL RESULTS

### 4.1 EXPERIMENTAL SETTINGS

During the training phase, we randomly crop the LR image to a size of 128×128. We train our models for 250K iterations with batch size of 4. We employ the Adam (Kingma & Ba, 2014) optimizer along with a cosine learning rate decay strategy. The learning rate decreases from $1 \times 10^{-4}$ to $1 \times 10^{-6}$. Additionally, we set the number of tokens in two dictionaries $N$ to 128, and initialize it using a normal distribution. We use PSNR, SSIM and LPIPS as performance metrics.

## 4.2 DATASETS

We conduct comparisons on three DuSR datasets (Yue et al., 2024), namely DuSR-Real, CameraFusion-Real, and RealMCVSR-Real. Each dataset contains complete triplets (LR, Ref, HR). DuSR-Real ($1792 \times 896$) has 420 triples for training and 55 for testing; CameraFusion-Real ($3584 \times 2560$) has 83 triples for training and 15 for testing; RealMCVSR-Real ($1792 \times 896$) has 330 triples for training and 50 for testing.

## 4.3 COMPARISONS WITH STATE-OF-THE-ART METHODS

We compare the proposed SDDuSR with three different super-resolution methods, including SISR methods: RCAN (Zhang et al., 2018), SwinIR (Liang et al., 2021), ESRGAN (Wang et al., 2018), BSRGAN (Zhang et al., 2021), RefSR methods: TTSR (Yang et al., 2020), MASA (Lu et al., 2021), DATSR (Cao et al., 2022), DuSR methods: DCSR (Wang et al., 2021), SelfDZSR (Zhang et al., 2022b), ZeDuSR (Xu et al., 2023), KeDuSR (Yue et al., 2024).

We evaluated SDDuSR on three datasets, as shown in Table 1. In the table, Full-Image represents the entire LR image, and Corner-Image represents the region outside the overlapping FoV. For models trained with only reconstruction loss, we denote them with the suffix '-rec'. On the three datasets, our method achieves SOTA performance on multiple metrics. SDDuSR has a greater lead on Corner-Image than on Full-Image. This is because the corner region of the LR image has low similarity with $I^{LRC}$, and warping operations alone cannot produce satisfactory results. The TDL strategy we proposed can achieve better results in this case. We have provided comparative results of the center region in the APPENDIX.

Figure 3 shows the visual comparisons on three DuSR datasets. In the center region, although we only performed feature matching on the high-frequency areas through SFM, there was no degradation in visual quality, and the texture details were still clearly restored. In the corner regions, due to the limitations of the warping operation, other methods may produce blurring or artifacts, while our method can effectively avoid these problems.

## 4.4 ABLATION STUDY

We conducted ablation studies on the proposed SFM and TDL strategies on the DuSR-Real dataset, as shown on the left side of Table 2.

When conducting ablation studies on SFM and TDL, we replaced SFM with conventional dense feature matching and removed TDL to serve as the baseline model. As can be seen from the table, when SFM is used to replace dense feature matching, the PSNR only drops by 0.01dB. This indicates that SFM does not lead to significant performance degradation and fully demonstrates that dense feature matching indeed introduces computational redundancy. Moreover, after adding the TDL strategy, the PSNR increased by 0.16dB, reaching the best performance of 27.81dB. These experiments clearly demonstrate the effectiveness of SFM and TDL.

We further investigated the impact of the number of tokens in the dictionary of TDL on the results, as shown on the right side of Table 2. We gradually increased $N$ from 64 to 256. As can be seen from the table, with the increase of $N$, the performance of the model even shows a downward trend. We believe there is a balance between $N$ and the scale of the dataset. When $N$ is small, the token dictionary cannot fully represent the features of the image. When $N$ is too large, due to the limited number of images in the dataset, the token dictionary will have redundant representations of features. Therefore, we set the number of tokens to 128. We have provided the visualization of TDL in the APPENDIX to better understand its function.

## 4.5 COMPUTATIONAL COMPLEXITY ANALYSIS OF SFM

We further analyzed the computational complexity of SFM on three datasets, as shown in Table 3. As can be seen from the table, compared with dense feature matching, the method proposed in this paper reduces the computational load by an average of 30%.

To more intuitively demonstrate the effect, we input $I^{LR}$ and $I^{LRC}$ into the mask generator and visualized the generated masks, as shown in Figure 4. In the masks, white indicates the high-frequency

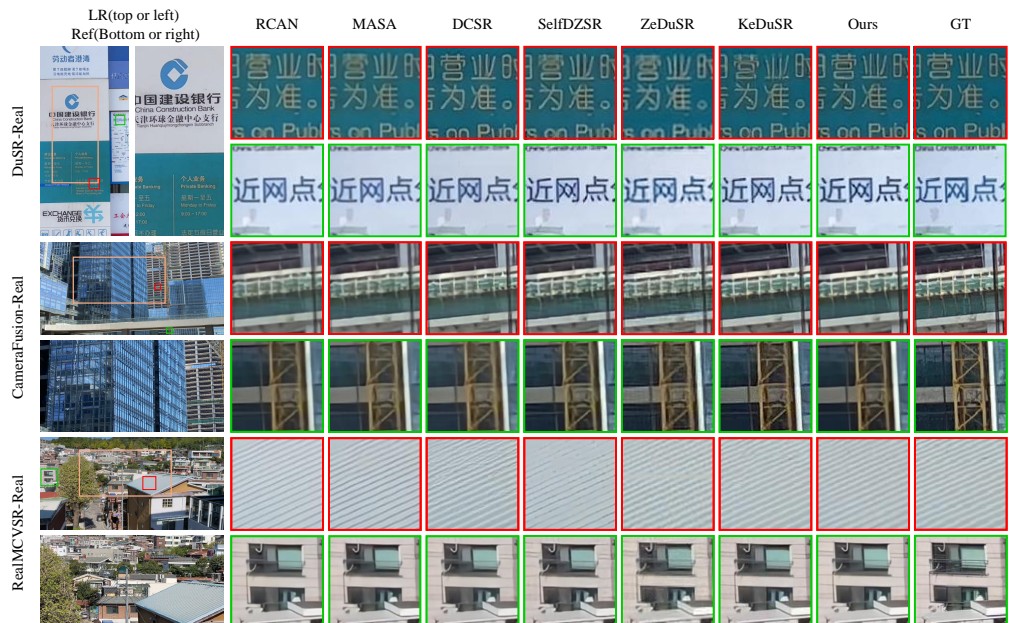

Figure 3: Visual comparisons on three DuSR datasets. The orange box indicates the overlapping FoV area, while the red and green boxes respectively represent the patches of the center region and the corner region. All results are obtained with only reconstruction loss and mask loss. (Zoom-in for best view)

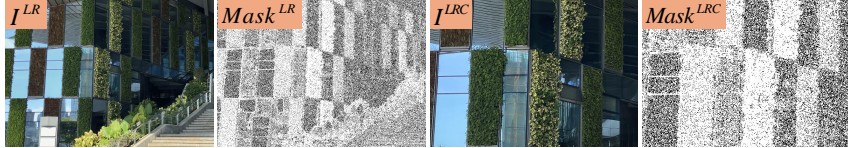

Figure 4: Visualization of masks. White indicates high-frequency regions, and black indicates low-frequency regions.

regions that need to be involved in feature matching, and black indicates the low-frequency regions that do not need to be involved in feature matching. As can be seen from the figure, SFM successfully distinguishes between the smooth (such as glass) and complex regions (such as the texture of leaves) of the two images. We only perform feature matching on the white areas of the two images to reduce computational complexity. We have provided more mask visualization results and a comparison of model parameters and inference speed in the APPENDIX.

## 5 CONCLUSION

In this paper, we propose a new Dual-Lens Super-Resolution method called SDDuSR. We analyze two key issues existing in current DuSR methods: (1) the smooth low-frequency regions in the LR image can achieve good visual effects without any reference, which leads to significant computational redundancy caused by dense feature matching, and (2) due to the inherent limitations of the warping operation, it is not possible to fully utilize the high-resolution features of the Ref image. To address these issues, we propose the SFM and TDL strategies. SFM separates the high-frequency and low-frequency regions of the images involved in feature matching through a mask generator and only performs feature matching between the high-frequency regions. SFM reduces the computational complexity by 30% while maintaining performance. TDL first defines two token dictionaries to avoid the resolution gap, and then updates the high-resolution features of the Ref image into the token dictionaries through cross-attention. Finally, the LR image interacts with the two token dictio-

naries through cross-attention to enrich the details. Extensive experiments have demonstrated that our method achieves SOTA results in the task of DuSR.

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

# A APPENDIX

## A.1 COMPARISON OF THE MODEL PARAMETERS AND INFERENCE SPEED

Table 4: Comparison of the model parameters and inference speed on DuSR-Real dataset. We evaluated all models on a single NVIDIA RTX 4070 GPU.

|  | SwinIR | ESRGAN | TTSR | MASA | DATSR | DCSR | SelfDZSR | KeDuSR | SDDuSR |
|---|---|---|---|---|---|---|---|---|---|
| Params (M) | 11.75 | 16.70 | 6.25 | 4.02 | 18.00 | 3.19 | 0.52 | 5.63 | 6.07 |
| Latency (s) | 6.41 | 0.18 | 16.90 | 3.42 | 21.04 | 1.89 | 0.38 | 1.14 | 1.21 |
| PSNR | 26.14 | 25.78 | 26.48 | 26.36 | 26.17 | 26.77 | 26.27 | 27.66 | 27.81 |

We evaluated the model parameters and inference speed of SDDuSR and other methods in Table 4. Our method is faster than SwinIR (Liang et al., 2021), TTSR (Yang et al., 2020), MASA (Lu et al., 2021), DATSR (Cao et al., 2022), DCSR (Wang et al., 2021). Compared with the current best method KeDuSR (Yue et al., 2024), our method has better performance with almost the same inference speed.

## A.2 QUANTITATIVE COMPARISONS OF FULL-IMAGE AND CENTER-REGION

Table 5: Quantitative comparisons of Full-Image and Center-Region

| Method | DuSR-Real | | CameraFusion-Real | | RealMCVSR-Real | |
|---|---|---|---|---|---|---|
|  | Full-Image | Center-Region | Full-Image | Center-Region | Full-Image | Center-Region |
|  | PSNR / SSIM / LPIPS | PSNR / SSIM | PSNR / SSIM / LPIPS | PSNR / SSIM | PSNR / SSIM / LPIPS | PSNR / SSIM |
| RCAN-rec | 26.44 / 0.8676 / 0.147 | 26.91 / 0.8704 | 25.67 / 0.8049 / 0.308 | 26.65 / 0.8158 | 25.96 / 0.8033 / 0.234 | 25.69 / 0.7937 |
| SwinIR-rec | 26.14 / 0.8601 / 0.157 | 26.35 / 0.8612 | 25.32 / 0.8007 / 0.315 | 25.81 / 0.8073 | 25.78 / 0.7982 / 0.246 | 25.50 / 0.7885 |
| ESRGAN | 25.78 / 0.8622 / 0.152 | 25.91 / 0.8637 | - | - | - | - |
| BSRGAN | 24.77 / 0.8227 / 0.202 | 25.01 / 0.8233 | - | - | - | - |
| TTSR-rec | 26.48 / 0.8676 / 0.147 | 27.69 / 0.8810 | 25.83 / 0.8044 / 0.311 | 26.75 / 0.8188 | 25.92 / 0.8017 / 0.235 | 25.94 / 0.7962 |
| MASA-rec | 26.36 / 0.8592 / 0.160 | 26.85 / 0.8620 | 25.78 / 0.8030 / 0.303 | 26.70 / 0.8155 | 25.95 / 0.7989 / 0.239 | 25.81 / 0.7899 |
| DATSR-rec | 26.17 / 0.8583 / 0.157 | 26.48 / 0.8596 | | | 25.81 / 0.7975 / 0.242 | 25.58 / 0.7882 |
| DCSR-rec | 26.77 / 0.8748 / 0.134 | 28.87 / 0.9078 | 26.02 / 0.8123 / 0.293 | 28.37 / 0.8440 | 26.28 / 0.8111 / 0.217 | 27.19 / 0.8298 |
| DCSR | 26.19 / 0.8553 / 0.110 | 28.05 / 0.8929 | 25.47 / 0.7605 / 0.165 | 27.14 / 0.7883 | 25.85 / 0.7966 / 0.186 | 26.98 / 0.8476 |
| SelfDZSR-rec | 26.27 / 0.8559 / 0.158 | 26.97 / 0.8591 | 25.94 / 0.8041 / 0.283 | 27.10 / 0.8148 | 25.33 / 0.7928 / 0.246 | 25.66 / 0.7860 |
| SelfDZSR | 25.98 / 0.8455 / 0.105 | 26.61 / 0.8496 | 25.64 / 0.7790 / 0.151 | 26.77 / 0.7897 | 25.24 / 0.7786 / 0.175 | 25.50 / 0.7732 |
| ZeDuSR-rec | 25.41 / 0.8247 / 0.191 | 26.29 / 0.8336 | 26.16 / 0.7920 / 0.279 | 27.44 / 0.8067 | 24.98 / 0.7702 / 0.262 | 25.38 / 0.7650 |
| KeDuSR-rec | 27.66 / **0.8890** / 0.117 | 29.58 / **0.9303** | 27.53 / **0.8292** / 0.276 | 30.48 / **0.8656** | 27.05 / **0.8406** / 0.180 | 29.25 / **0.9191** |
| KeDuSR | 27.18 / 0.8752 / **0.084** | 29.06 / 0.9219 | 27.00 / 0.7931 / **0.133** | 29.77 / 0.8418 | 26.42 / 0.8184 / **0.127** | 28.51 / 0.9090 |
| SDDuSR-rec | **27.81** / 0.8874 / 0.121 | 29.60 / 0.9230 | **27.60** / 0.8274 / 0.280 | 30.21 / 0.8594 | **27.09** / 0.8386 / 0.183 | **29.26** / 0.9135 |
| SDDuSR | 27.20 / 0.8722 / **0.084** | 28.99 / 0.9176 | 27.02 / 0.7897 / 0.135 | 29.47 / 0.8309 | 26.71 / 0.8290 / 0.151 | 28.65 / 0.9029 |

Table 5 shows the comparative results of the center region. From the table, it can be seen that the performance of the center region has not significantly improved and is even weaker than KeDuSR Yue et al. (2024) on CameraFusion-Real dataset. This is because CameraFusion-Real dataset has a higher resolution, and the number of patches that did not participate in matching during the SFM stage is greater than other datasets. SDDuSR aims to improve the performance of corner region through TDL, although the performance improvement in the center region is not good, it is higher than other methods on the entire image.

## A.3 ROBUSTNESS EVALUATION

To evaluate the robustness of our method on untrained datasets, we evaluated the model trained only on the DuSR-Real dataset on two other datasets, the experimental results are shown in Table 6. It can be seen that SDDuSR still has the best performance on untrained datasets, which also proves that SDDuSR has good robustness.

## A.4 MORE VISUALIZATION RESULTS

Figure 6 shows the visualization results of TDL. We present the comparison results between the warping feature $F^{Warp}$ and the dictionary feature $F^{Dict}$. In Figure 6, the red rectangular box area

Table 6: Robustness evaluation with the model trained on DuSR-Real.

| Method | RealMCVSR-Real PSNR / SSIM | CameraFusion-Real PSNR / SSIM |
|---|---|---|
| TTSR-rec | 24.67 / 0.7814 | 25.23 / 0.7760 |
| MASA-rec | 24.99 / 0.7830 | 25.45 / 0.7769 |
| DCSR-rec | 25.46 / 0.7986 | 25.58 / 0.7931 |
| SelfDZSR-rec | 24.86 / 0.7778 | 25.55 / 0.7805 |
| ZeDuSR-rec | 24.98 / 0.7702 | 26.16 / 0.7920 |
| KeDuSR-rec | 26.55 / **0.8325** | 27.24 / **0.8178** |
| SDDuSR-rec | **26.64** / 0.8272 | **27.25** / 0.8126 |

in the LR image contains the features of leaves, but it is not present in the Ref image. In this case, still using feature matching and warping operations inevitably introduces irrelevant noise, so the warping features $F^{Warp}$ visualized in Figure 6 are very messy.

In contrast, the dictionary in TDL is continuously updated during the training phase, obtaining the feature representation of the entire dataset. Therefore, for the red rectangular box area, TDL obtained a better feature representation by looking up the dictionary, avoiding the introduction of irrelevant features.

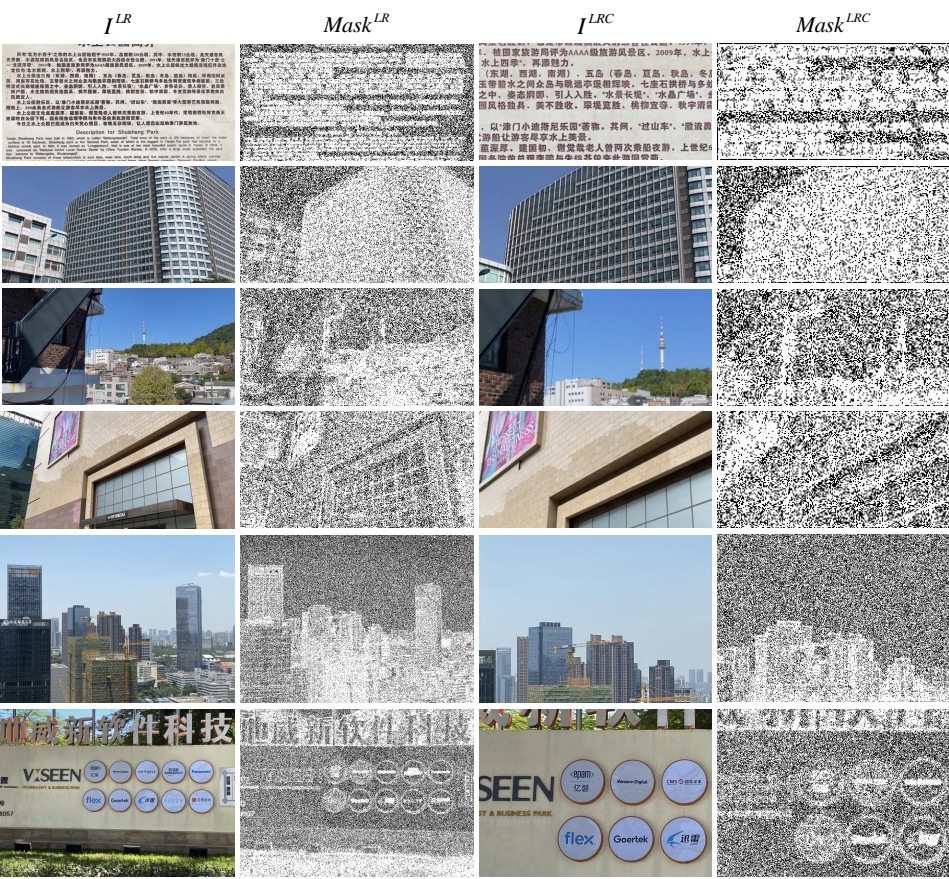

Figure 5: Mask visualization results on three datasets. The first two rows are for the DuSR-Real dataset, the middle two rows are for the RealMCVSR-Real dataset, and the last two rows are for the CameraFusion-Real dataset.

In order to better understand the semantic information represented by different tokens in the dictionary, we visualized them, as shown in Figure 7. We perform *argmax* operation on the attention map $A_{up}$ obtained from Equation 6 to obtain the most relevant token for each pixel in the image, and visualize it as a binary image. From the figure, it can be seen that different tokens represent

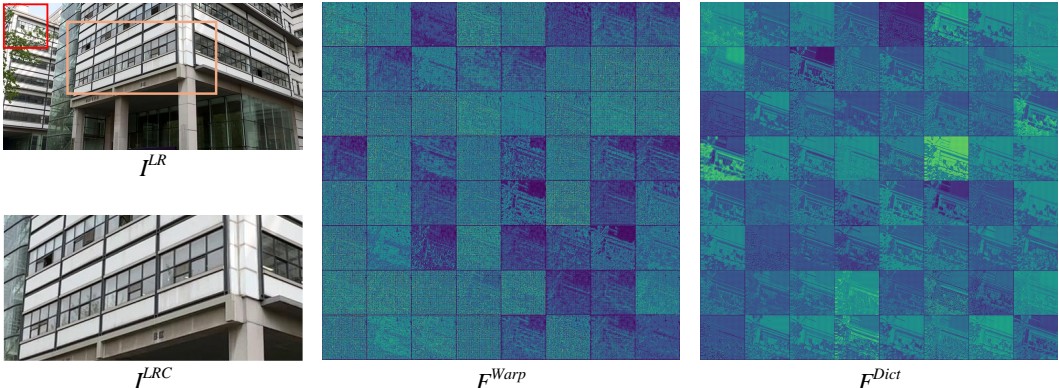

Figure 6: The visualization results of TDL. The red box represents the visualization area of the two columns on the right.

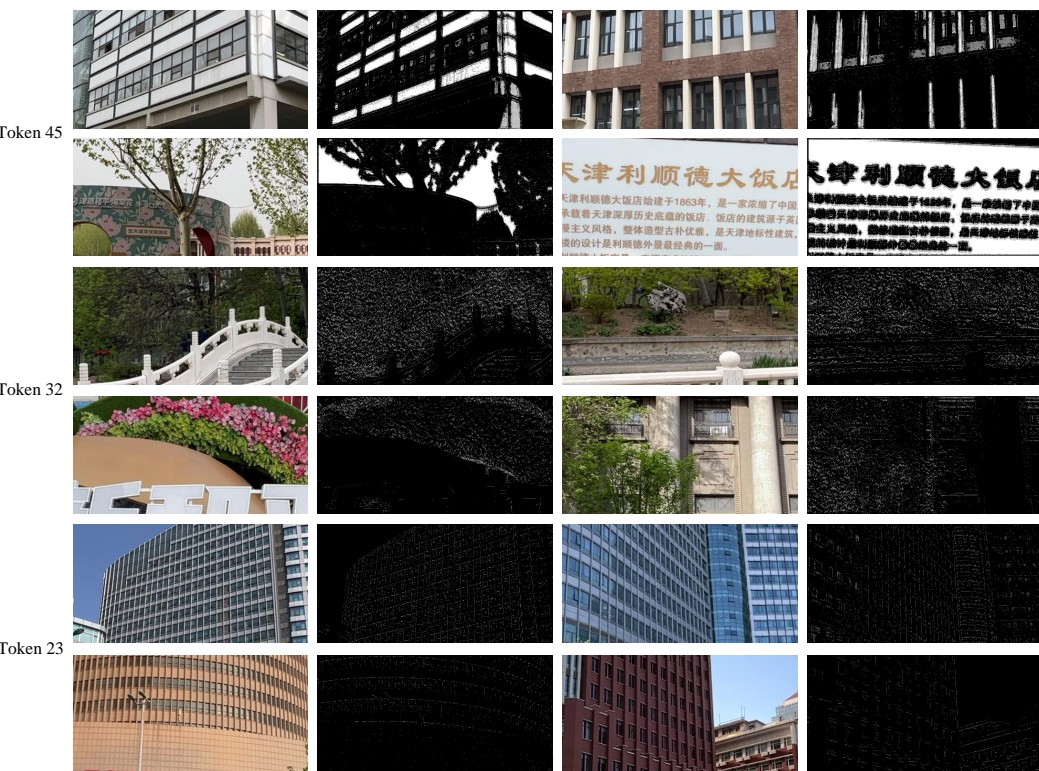

Figure 7: Visualization of semantic information represented by different tokens in the dictionary.

different semantic information. The 45th token represents light colored low-frequency features, the 32nd token represents complex texture features of leaves, and the 23rd token represents regular edge features.

### A.5 THE USE OF LARGE LANGUAGE MODELS (LLMS)

A large language model (ChatGPT-4, OpenAI) was used solely to polish the English prose of this paper. The model was engaged only for grammar, style, and wording improvements; it was not involved in generating any scientific content, ideas, experimental designs, results, or interpretations. All authors retain full responsibility for the final text and for the scientific accuracy of the work.

