# OpenReview forum: "SDDuSR: Sparse Feature Matching and Token Dictionary Learning for  Dual-Lens Super-Resolution"
_ICLR.cc/2026/Conference — ICLR 2026 Conference Withdrawn Submission_

### Official Review · Reviewer_4EAa · 2025-10-30

**Soundness:** 3
**Presentation:** 3
**Contribution:** 3
**Rating:** 6
**Confidence:** 3

**Summary:**

The paper proposes SDDuSR, a Dual-Lens Super-Resolution method that improves efficiency and feature utilization compared to existing DuSR approaches. Instead of dense feature matching and warping, the method performs sparse feature matching only in high-frequency regions using a mask generator, and leverages token dictionary learning to better transfer high-resolution information from the reference image. Experiments show that SDDuSR achieves superior quantitative and qualitative performance over state-of-the-art methods.

**Strengths:**

The paper proposes a well-motivated combination of sparse feature matching and token dictionary learning, which effectively addresses both the computational redundancy and feature utilization limitations in existing DuSR approaches.
The paper integrates several components (mask generator, flow-guided alignment, deformable convolution, and token dictionary learning) in a coherent and technically sound framework. The method is rigorously designed, with strong mathematical formulation and empirical validation.

**Weaknesses:**

While the overall framework is interesting, several core components are standard techniques. The originality mainly lies in their integration rather than in new algorithmic contributions.

**Questions:**

1.	While the effect of token number N is explored, the paper does not analyze why larger dictionaries degrade performance or what semantic structures tokens capture.
2.	Have the authors analyzed what patterns or structures the token dictionaries learn? Visualizing token attention or diversity could enhance understanding.
3.	How does SDDuSR perform under misaligned or low-light conditions? Could the authors discuss the robustness of SFM and TDL in such scenarios?

---

> ### Author Response · Authors · 2025-11-18
> **Rebuttal by Authors**
>
> **Weaknesses:**
>
> For SFM, we first pointed out and verified the prior in DuSR that "low-frequency regions do not require reference", and based on this, introduced the idea of mask sparsity into feature matching. Although some previous works have proposed mask sparsity in image classification, due to the particularity of DuSR, there is a problem of gradient backpropagation in the feature matching stage. As mentioned in section 3.2 of the paper, simply introducing mask sparsity will result in insufficient training. Therefore, we also converted the warping operation into matrix multiplication through the STE strategy, which is the first sparse matching process in the DuSR field.
>
> For TDL, we are the first to point out the limitations of warping operations, which are not present in other DuSR and RefSR methods. Although dictionaries have been widely used in some fields such as image classification, they cannot be directly introduced into the DuSR field. As mentioned in section 3.3 of the paper, there is a resolution gap in the DuSR task, so we set up a dual dictionary and synchronous update mechanism to address this issue. We are the first to propose using dictionary learning to address the limitations of warping operations in DuSR tasks.
>
> **Questions:**
>
> **1.** In TDL, a larger N is not necessarily better. Due to the limitation of the number of images in the dataset, an excessively
> large N can cause redundancy in the representation ability of the dictionary. We hope that each token in the dictionary
> represents different semantics. If N is too large, it will result in multiple tokens representing the same semantics, leading to
> performance degradation. In addition, we visualized the semantics represented by each token in the APPENDIX,
> as well as the comparison between the warping feature $F^{Warp}$ and the dictionary feature $F^{Dict}$
>
> **2.** We visualized the semantics represented by each token in the APPENDIX.
>
> **3.** Thank you for this suggestion. As DuSR is a relatively novel task, there are currently no other datasets under specific conditions for us to evaluate.

---

> > ### Comment · Reviewer_4EAa · 2025-11-26
> >
> > The authors’ responses have addressed most of the concerns raised. Regarding the last question, in the absence of a dedicated dataset, it would be helpful if the authors could analyze failure cases and their causes based on the existing experimental datasets. This could provide additional insights into the model’s robustness.

---

> > > ### Author Response · Authors · 2025-11-27
> > >
> > > Thank you for your suggestion. Due to the lack of misaligned and low-light datasets for DuSR tasks. Therefore, we only trained the model on the DuSR-Real dataset and then tested it on two other datasets to evaluate the robustness of the model on untrained datasets. The experimental results are supplemented in Table 6 of the APPENDIX.

---

> ### Author Response · Authors · 2025-11-26
>
> Dear Reviewer,
>
> I hope this message finds you well. As the discussion period is nearing its end, l wanted to ensure we have adressed all your concerns satisfactorily. If there are any additional points or feedback you'd like us to consider, please let us know. Your insights are invaluable to us, and we're eager to address any remaining issues to improve our work.
>
> Thank you for your time and effort in reviewing our paper.

---

### Official Review · Reviewer_1p2t · 2025-10-31

**Soundness:** 3
**Presentation:** 3
**Contribution:** 2
**Rating:** 4
**Confidence:** 5

**Summary:**

This paper addresses two major challenges in dual-camera super-resolution (DuSR): computational redundancy and insufficient utilization of reference features. The authors propose an innovative framework called SDDuSR, which integrates Sparse Feature Matching (SFM) and Token Dictionary Learning (TDL). The SFM module employs a mask generator to dynamically identify high-frequency detail regions and performs feature matching only within these regions, thereby reducing computational complexity. The TDL module maintains an online-updatable token dictionary, enabling the model to overcome the rigidity of conventional deformation operations and to reconstruct fine details from richer feature priors.

**Strengths:**

1. The SFM module introduces a novel perspective for DuSR tasks — focusing on high-frequency regions only, without causing significant performance degradation.
2.The paper is well-structured, clearly written, and the formulas are easy to follow.

**Weaknesses:**

1. The novelty is incremental. The ideas of SFM and TDL are not new strategies.
2 The experimental results are not convincing. One of the core contributions of this paper is using sparse feature matching to reduce dense computation in the spatial dimension. However, the proposed TDL itself is dense computation. Specifically, in the learning stage of TDL, each low-resolution feature vector needs to calculate the similarity with every token in the dictionary to produce an attention map. This is essentially a kind of global dense matching in the feature dictionary dimension, and the redundancy problem does not seem to be solved. It may even make the proposed method less efficient than traditional dense matching methods.
3. It is suggested to compare the overall computational efficiency of the model with other methods, such as KeDuSR, to prove the efficiency of the proposed model, instead of only discussing the efficiency gain brought by the SFM module. In addition, adding more metrics such as runtime and memory usage can more comprehensively demonstrate the efficiency of SDDuSR .
4. It is suggested to analyze the sparsity of the mask generated in SDDuSR-rec, which corresponds to using only the reconstruction loss Lres. Because without the Lmask constraint, the mask values may become all 1, which would degenerate into dense feature matching, and SFM would lose its essential function.
5. In Fig. 2(a), ILR and IRef are the wide-angle image and the telephoto image, respectively, but the two images have the same field of view (FOV), which is confusing.
6The mask generator of SFM relies on a threshold, yet it’s unclear how robust this heuristic is across different datasets or content types.
7.Relative to KeDuSR, SDDuSR lacks explicit quantitative comparison on Center-Image regions (Table 1 reports partial metrics for KeDuSR but lacks direct Center-Image PSNR/SSIM comparisons with SDDuSR).
8.Compared with KeDuSR, the visual improvement of SDDuSR is marginal and occasionally exhibits detail smoothing, which contradicts the claimed effectiveness of TDL in enriching texture details.

**Questions:**

1.What is the quantitative statistics on the mask generator output, such as mean/variance of high-frequency regions kept per image for various datasets and settings? Or can the authors provide the results compared to other adaptive modules?
3.What is the quantitative comparison on Center-Image regions?
4.Why are there the observed detail smoothing phenomenon and  why does TDL fail to synthesize sharper textures in these cases?
Other question see Weakness 2,3,4.

---

> ### Author Response · Authors · 2025-11-18
> **Rebuttal by Authors**
>
> **Weaknesses:**
>
> **1.** Thank you for this suggestion. For SFM, we first pointed out and verified the prior in DuSR that "low-frequency regions do not require reference", and based on this, introduced the idea of mask sparsity into feature matching. Although some previous works have proposed mask sparsity in image classification, due to the particularity of DuSR, there is a problem of gradient backpropagation in the feature matching stage. As mentioned in section 3.2 of the paper, simply introducing mask sparsity will result in insufficient training. Therefore, we also converted the warping operation into matrix multiplication through the STE strategy. We are the first to propose sparse matching in the DuSR task
>
> For TDL, we are the first to point out the limitations of warping operations, which are not present in other DuSR and RefSR methods. Although dictionaries have been widely used in some fields such as image classification, they cannot be directly introduced into the DuSR field. As mentioned in section 3.3 of the paper, there is a resolution gap in the DuSR task, so we set up a dual dictionary and synchronous update mechanism to address this issue. We are the first to propose using dictionary learning to address the limitations of warping operations in DuSR tasks.
>
> In addition, in TDL, we set the number of tokens N of the dictionary to be much smaller than the resolution of the image, which does not bring significant computational burden. We provided the parameter count and inference speed of the entire model in the APPENDIX.
>
> **2.** We have added a comparison of parameters and inference speed in the APPENDIX.
>
> **3.** This is our negligence. We trained two models, SDDusR-rec and SDDusR, respectively. Among them, SDDuSR-rec uses reconstruction loss and mask loss for training, while SDDuSR uses all losses for training, not just the results of training with reconstruction loss. We have provided explanations in section 3.4 of the paper and at the header of Table 1.
>
> **4.** This is our negligence, we have already made revisions in the paper.
>
> **5.** As DuSR is a relatively novel task, there are currently no other datasets under specific conditions for us to evaluate. Therefore, we only trained the model on the DuSR-Real dataset and then tested it on two other datasets to evaluate the robustness of the model on untrained datasets. The experimental results are supplemented in Table 6 of the APPENDIX.
>
> **6.** We have provided comparative results of the center region in the APPENDIX.
>
> **7.** We understand your concerns. When there are no similar features to LR images in Ref images, TDL can enrich the details of some dissimilar areas (such as the first, second, and fifth rows in Figure 3), and eliminate the artifacts caused by warping operations (such as the third, fourth, and sixth rows in Figure 3). In addition, we provide a visual comparison between the warping feature A and the dictionary feature B in the APPENDIX.
>
> **Questions:**
>
> **1.** I'm very sorry, I may not quite understand what you mean. SFM directly outputs binary masks through Gumbel Softmax, and the entire training process is dominated by reconstruction loss, with mask loss as the constraint term, and the model autonomously learns to distinguish between high and low frequency regions. What specific adaptive modules do you refer to?
>
> **2.** We have provided comparative results of the center region in the APPENDIX.

---

> ### Author Response · Authors · 2025-11-26
>
> Dear Reviewer,
>
> I hope this message finds you well. As the discussion period is nearing its end, l wanted to ensure we have adressed all your concerns satisfactorily. If there are any additional points or feedback you'd like us to consider, please let us know. Your insights are invaluable to us, and we're eager to address any remaining issues to improve our work.
>
> Thank you for your time and effort in reviewing our paper.

---

### Official Review · Reviewer_43qp · 2025-11-01

**Soundness:** 3
**Presentation:** 3
**Contribution:** 2
**Rating:** 4
**Confidence:** 4

**Summary:**

This paper proposes a Dual-Lens Super-Resolution (DuSR) framework named SDDuSR, which integrates Sparse Feature Matching (SFM) and Token Dictionary Learning (TDL) to address two major limitations of existing DuSR methods: computational redundancy in dense matching and incomplete utilization of reference features. Specifically, a mask generator distinguishes high- and low-frequency regions, allowing feature matching only in high-frequency areas to reduce redundant computation. Meanwhile, the TDL module abstracts reference features into token dictionaries and employs cross-attention to transfer high-resolution priors to the low-resolution image. Experiments on multiple real-world DuSR datasets show its effectiveness to some extent.

**Strengths:**

The paper is well-motivated. The proposed SFM effectively reduces redundant computation by focusing on high-frequency regions without performance degradation, while the TDL module captures higher-level semantic priors and overcomes the rigidity of traditional warping operations.

**Weaknesses:**

1. In Section 3.1, the fusion of SFM and TDL features relies on simple concatenation and convolutional fusion; more interpretable or adaptive fusion strategies could be explored.

2. In Section 3.3, the TDL design lacks a detailed analysis of key parameters such as update frequency, and attention layer depth.

3. The experiments mainly report PSNR, SSIM, and LPIPS metrics, but do not provide efficiency indicators such as runtime, parameter count, or inference speed.

4. In Section 4.5, the experiment only demonstrates the computational savings of SFM relative to DFM, while omitting the computational cost of the TDL module. To substantiate the claim that "the overall computing load is almost unchanged," it is recommended to provide the total GFLOPS or latency for the entire SDDuSR model (which includes TDL) and present a fair comparison against SOTA methods like KeDuSR.

5. In Section 3.1, the text states, “In the updating phase, the features of F^LRC and F^Refare updated into D^LRC and D^Ref”，however, according to Fig. 2 and the description in Section 3.3, F^Ref should be F_align^Ref.

6. In 3 Methodology, there are too many abbreviations and symbol definitions, which negatively affect the readability.

**Questions:**

1.Will the mask generator misclassify complex texture regions, thereby causing the omission of high-frequency features?

2.SFM reduces the computational load by about 30%, but is the actual speed improvement consistent across GPU platforms?

3.Could more visualizations (such as mask heatmaps and attention distributions of the feature dictionary) be added to aid understanding?

---

> ### Author Response · Authors · 2025-11-18
> **Rebuttal by Authors**
>
> **Weaknesses:**
>
> **1.** A better fusion strategy can indeed further improve performance, but initially we hoped that the fusion strategy would be consistent with the baseline model KeDuSR. We will further explore more effective fusion strategies in future research.
>
> **2.** In this paper, TDL does not require stacking depth or setting the number of heads like Transformer based SR methods, but only performs cross attention once. In addition, the number of updates to the dictionary is the same as the number of iterations during the entire model training phase. Each iteration updates the information of the batch of images to the dictionary and controls the content through parameter s, ultimately abstracting the features of the entire dataset into the dictionary.
>
> **3.** We have added a comparison of parameters and inference speed in the APPENDIX.
>
> **4.** We have added a comparison of parameters and inference speed in the APPENDIX.
>
> **5.** This is our negligence, we have already made revisions in the paper.
>
> **6.** We have carefully reviewed and evaluated the entire text again in response to your issue of "too many symbols defined in the paper". Due to the involvement of multiple inputs in this paper, in order to ensure the accuracy and reproducibility of the narrative, we have to introduce a certain number of symbols in the main text. After careful consideration, the symbols currently retained are directly related to subsequent derivations, and further deletion may affect the integrity of the paper.
>
> **Questions:**
>
> **1.** This is possible, as mentioned in section 3.4, we hope the model can autonomously decide which regions are high-frequency regions. Therefore, if certain high-frequency regions are missed, it indicates that the matching results for that region will not affect the final performance.
>
> **2.** We have added a comparison of parameters and inference speed in the APPENDIX.
>
> **3.** We have added the visualization results of TDL in the APPENDIX to further assist in understanding. For masked heatmaps, since Gumbel Softmax directly outputs binary results (pixel values of 0 or 1), Figure 4 is the output result of Gumbel Softmax. For better understanding, we have provided more masks and TDL visualization results in the APPENDIX.

---

> ### Author Response · Authors · 2025-11-26
>
> Dear Reviewer,
>
> I hope this message finds you well. As the discussion period is nearing its end, l wanted to ensure we have adressed all your concerns satisfactorily. If there are any additional points or feedback you'd like us to consider, please let us know. Your insights are invaluable to us, and we're eager to address any remaining issues to improve our work.
>
> Thank you for your time and effort in reviewing our paper.

---

### Note · Authors · 2026-01-30

I have read and agree with the venue's withdrawal policy on behalf of myself and my co-authors.

---

### Meta-Review · Area_Chair_4fNQ · 2026-01-04

**Summary:**

The paper proposes SDDuSR, a method for Dual-Lens Super-Resolution that combines a Sparse Feature Matching (SFM) strategy to reduce computation in low-frequency regions with a Token Dictionary Learning (TDL) module to capture reference features.

While the problem of computational redundancy in Reference-based SR is valid, the consensus among the reviewers is that the proposed solution offers only incremental technical novelty. The method largely assembles existing standard techniques (masking based on frequency, dictionary learning via cross-attention) rather than introducing a significant algorithmic breakthrough. Despite the authors' efforts during the rebuttal to demonstrate efficiency gains, the fundamental concerns regarding the "marginal" visual improvements and the derivative nature of the core modules remain the primary drivers for this recommendation.

**Reviewer Concerns:**

Outstanding Concerns (Basis for Rejection):

(1) A shared concern (particularly from Reviewers 1p2t and 4EAa) is that the core components are standard. SFM is essentially a threshold-based masking operation, and TDL is a standard cross-attention mechanism. The integration of these existing blocks is logical but does not represent the level of innovation typically expected at ICLR. The rebuttal clarified the implementation but did not refute the critique that the underlying algorithms are standard.

(2) There is marginal visual improvement. Reviewer 1p2t pointed out that the visual improvement over SOTA methods (like KeDuSR) is marginal and, in some cases, exhibits detail smoothing. While the authors argued that TDL helps in corner cases, the lack of a decisive qualitative advantage makes it difficult to justify the complexity of the proposed dictionary mechanism, regardless of the efficiency claims.

(3)  There remains a conceptual disconnection regarding the TDL module. As noted by Reviewer 1p2t, while SFM is spatially sparse, the TDL module involves global dense matching against the dictionary tokens. The authors' defense—that the number of tokens is small ($N=128$)—addresses the computational cost but not the theoretical inconsistency of the "sparse" framework.

(4)  Reviewer 43qp noted that the fusion of features is simplistic (concatenation). The authors acknowledged this and deferred it to future work, but for a conference paper, this lack of architectural maturity is a drawback.

Concerns Addressed:

The initial lack of runtime and parameter comparisons (raised by Reviewers 43qp and 1p2t) was addressed in the appendix during the rebuttal. The data confirms the method is efficient, but efficiency alone does not rescue the paper from the novelty concerns.

**Reviewer Scores:**

Reviewer 43qp (Score: 4 -> 4/6): This reviewer identified the contribution as "fair" and criticized the fusion strategy. Even with the added efficiency stats, the fundamental critique of the method's depth likely keeps this score at a 4 or weak 6.

Reviewer 1p2t (Score: 4 -> 4): This reviewer was the most critical regarding novelty and visual quality (smoothing artifacts). The rebuttal addressed robustness but did not fundamentally change the architecture or the visual output. Their score is likely to remain a 4 (Reject).

Reviewer 4EAa (Score: 6 ->6/4): While this reviewer was more positive about the integration, they also conceded that the components are "standard techniques." In light of the other reviewers' stronger critiques on novelty and visual marginality, their enthusiasm would likely dampen in a full committee discussion, potentially dropping to a 4 (Weak Reject).

---

### Decision · Program_Chairs · 2026-01-26

Reject